# MIPGen: Learning to Generate Scalable MIP Instances

## Abstract

Large-scale Mixed-Integer Programming (MIP) problems have been efficiently addressed using Machine Learning (ML)-based frameworks to obtain high-quality solutions. When addressing real-world MIP problems, ML-based frameworks often face challenges in acquiring sufficient isomorphic instances for practical training. This underscores the need for generators that can autonomously produce isomorphic MIP problems from existing instances. This paper introduces MIPGen, a novel generative framework for autonomous MIP instance generation. Our key contribution lies in the three-stage problem generation in MIPGen: 1) Instances Classification, which learns and clusters the embeddings of a bipartite graph representation of the problem; 2) Node Splitting and Merging, which splits the bipartite graph and tries to reconstruct it; 3) Scalable Problem Construction, which concatenates tree structures to get larger problems. We demonstrate that the instances generated by MIPGen are highly similar to the original problem instances and can effectively enhance the solution effect of the ML-based framework. Further experiments show that the scaled-up generated instances still retain the problem's structural properties, validating the proposed framework's effectiveness.

## 1 Introduction

Mixed-integer linear programming (MIP) is an extension of linear programming that addresses problems where at least one variable assumes a discrete integer value instead of a continuous one (Wolsey, 2007). For large-scale MIP problems, leveraging Machine Learning (ML)-based frameworks to find high-quality solutions has become increasingly popular due to its capacity to strike a balance between solution time and solution quality compared with traditional solution methods (Nair et al., 2020; Ye et al., 2023). These frameworks are usually trained on the same type of problems and can outperform traditional solvers on these types of problems. A significant challenge faced by these ML-based frameworks is the heavy reliance on a large number of isomorphic problem instances for training. Nevertheless, it's worth noting that numerous datasets (Koch et al., 2011; Gleixner et al., 2021) suffer from a shortage of such isomorphic instances. This underscores the requirement for a generator capable of autonomously producing isomorphic MIP instances from existing instances.

Current generators used to generate problem instances from existing instances can be roughly divided into two categories, mathematically constructed and ML-based approaches. Traditional methods for problem generation typically involve a manual analysis of dataset structures and mathematical formulations to create new isomorphic instances (Eichfelder et al., 2023). This approach heavily relies on manual design and often fails to accommodate a diverse range of problems. In contrast, ML-based approaches, especially graph neural network (GCN)-based methods have gained popularity for generating imitative instances. For instance, G2SAT (You et al., 2019) utilizes GCNs to generate new isomorphic instances of the Boolean Satisfiability Problem (SAT), marking a significant advancement in the domain. Nevertheless, these methods are limited to coefficient-free SAT problems and struggle to generate intricate MIP instances with specific coefficients and constraints.

To address the limitations of current generative methods and autonomously produce high-quality isomorphic MIP instances, this paper introduces MIPGen (Mixed Integer Programming Instance Generator), a deep generative framework designed for large-scale MIP instances. Inspired by extant generation strategies (You et al., 2019; Steever et al., 2022), the key points of MIPGen lies in three components: *Instances Classification*, *Node Splitting and Merging*, and *Scalable Problem Con-*

*struction*. When classifying instances, MIPGen first adopts a bipartite graph representation (BGR) combined with a random-feat strategy (Chen et al., 2022) to achieve an efficient and lossless feature embedding. Then, based on the BGR, MIPGen integrates a clustering approach to distinguish between different types of problems so that models can be trained separately for each type of problem. Subsequently, utilizing the isomorphic dataset obtained from instance classification, MIPGen integrates a discriminator model with GCN and MLP to split and merge nodes. In particular, it initially decomposes the original bipartite graph into tree-like structures, simultaneously collecting training data throughout the process. Subsequently, MIPGen utilizes the discriminator model to predict the graph structure of MIP instances. Furthermore, MIPGen scales problems by concatenating various tree structures of the same category to construct scalable problems.

Experimental results on three standard MIP demonstrate that MIPGen can proficiently generate MIP problems that resemble input training problems. By producing high-fidelity data imitations, it addresses the challenge of ML-based frameworks (Ye et al., 2023) depending on a multitude of isomorphic problem instances. This paper's contributions can be summarized as follows:

1. To the best of our knowledge, this is the first paper to propose a deep generative model designed to generate isomorphic MIP instances, laying the foundation for introducing large model pre-training to combinatorial optimization problems.

2. We introduce a method based on VGAE and EM algorithm for precise and efficient instance classification using a bipartite graph representation.

3. We automate node splitting operations to attain a simplified tree-based representation of the original problem. Moreover, we leverage a merging decision-maker grounded in GCN with half-convolutions to create isomorphic scalable MIP problems.

4. We show that the instances generated by MIPGen are highly similar to the original problem instances and can enhance the solution effect of the ML-based framework. The scaled-up generated instances still retain the problem's structural properties with improved solving difficulty.

## 2 PRELIMINARIES

**Bipartite Graph Representation.** As a lossless representation for MIP instances (Gasse et al., 2019), Bipartite Graph Representation (BGR) seamlessly transforms MIP instances into bipartite graphs suitable for GCN inputs as illustrated in Figure 1. The bipartite structure consists of $n$ decision variable nodes on the left and $m$ linear constraint nodes on the right. An edge $(i, j)$ with weight $a_{ij}$ denotes the participation of the $i$-th decision variable in the $j$-th constraint. Traditional feature selection in such representations only relies on types and coefficients, such as variable type and its coefficient in the objective function, and constraint types. However, some studies identify this may lead to a significant decline in the embedding ability with this approach, especially for "foldable" MIP instances (Chen et al., 2022). To improve this, the random feat strategy is incorporated into feature selection (Chen et al., 2022).

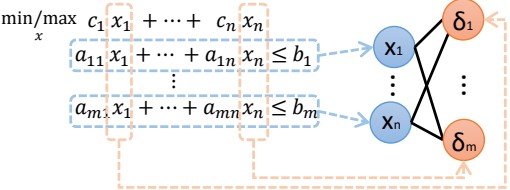

Figure 1: A bipartite graph of a MIP.

**Graph Convolutional Network.** Graph Convolutional Networks (GCNs) (Zhou et al., 2020) are frequently employed in BGR (Li et al., 2021; Cao et al., 2021). A typical GCN is defined as Equation 1, where $H^{(l)}$ is the node feature matrix at layer $l$; $A$ and $D$ are the adjacency matrix and the degree matrix of the graph respectively; $W^{(l)}$ is the weight matrix for layer $l$; and $\sigma$ is an activation function.

$$H^{(l+1)} = \sigma\left(D^{-\frac{1}{2}} A D^{-\frac{1}{2}} H^{(l)} W^{(l)}\right). \tag{1}$$

When processing data between two distinct entities, bipartite-based GCNs efficiently encode relationships which minimizes redundancy. By focusing on one half of the bipartite graph, half-convolution GCNs optimize computational efficiency, ensuring separate node set features are distinctively processed (Gasse et al., 2019; Yoon, 2022).

**Variational Graph Auto-Encoder.** Variational Graph Auto-Encoder (VGAE) addresses graph-structured data (Kipf & Welling, 2016), which is widely used in various graph analysis tasks such as node clustering (Mrabah et al.), community detection (Salha-Galvan et al., 2022), and link prediction (Bose et al., 2019). Formally, the loss calculation process of VGAE is defined as Equation 2, where $\mathcal{L}$ is the overall loss function to be minimized, combining the reconstruction loss and the Kullback-Leibler (KL) divergence between the approximate posterior and the prior.

$$\mathcal{L} = -\sum_{i,j} \left( A_{ij} \log(\hat{A}_{ij}) + (1 - A_{ij}) \log(1 - \hat{A}_{ij}) \right) + \mathrm{D_{KL}} \left( q_\phi(Z|X) \| p(Z) \right). \tag{2}$$

VGAE employs a GCN-based encoder to project the graph into a latent space, followed by a decoder that reconstructs the graph from this compressed representation. The variational aspect ensures that the model learns a probabilistic mapping, capturing uncertainties and providing a regularized latent space representation.

**Specific Graph Structure Generation.** The task of generating specific graph structures has grown in prominence with the rising significance of graph-based learning and applications. Some representative work includes Recurrent Neural Network (RNN) based framework (You et al., 2018), Generative Adversarial Network (GAN) based generation model (Wang et al., 2018) and transformer-based approach (Yun et al., 2019). As the deep generative framework to generate SAT formulas, G2SAT (You et al., 2019) utilizes Latent Convolutional Graphs to represent these SAT formulas and reframe SAT formula creation as a bipartite graph generation issue. The pivotal breakthrough of G2SAT is the identification of a process for generating any bipartite graph by starting with a set of trees and then applying sequential node merging operations on nodes from one of the two partitions. As nodes are merged, the initial trees combine, resulting in increasingly complex bipartite structures. The researchers also introduce the concept of node splitting as the inverse of node merging.

## 3 METHODOLOGY

This section introduces the proposed MIPGen (Mixed Integer Programming Instance Generator). Our generator consists of three key stages: instance classification (Sec. 3.1), Node Splitting and Merging (Sec. 3.2), and Scalable Problem Construction (Sec. 3.3). The whole framework of MIP-Gen is comprehensively illustrated in Figure 2, where the MIPGen pipelines are separated into *training steps* and *testing steps*. *Training steps* are active only during classifying the input MIP problem dataset and training the discriminator model to reconstruct the graph structure. *Testing steps* are active during scaling and generating MIP instances, incorporating the evaluation of generated instances.

### 3.1 INSTANCES CLASSIFICATION

To initiate the process with a set of *MIP instances*, the primary objective of MIPGen is to classify and group these instances into distinct categories. Such categorization is crucial as it forms the basis for the subsequent learning and generation phases, each tailored to a specific problem type. To achieve this, MIPGen represents problems as bipartite graphs, as described in Figure 2. Subsequently, it employs a Variational Graph Auto-Encoder (VGAE) to extract embeddings of both decision variables and constraints. This process yields node embeddings for each bipartite graph. Finally, MIPGen leverages the Expectation-Maximization (EM) algorithm, working in tandem with a Gaussian Mixture Model to segregate the various MIP instances into distinct clusters.

**Formal Representation of Existing Instances.** In addition to leveraging traditional bipartite graph representations, our MIPGen framework incorporates a novel random feat-based method to enhance its expressive power (Chen et al., 2022). Formally, let's define $h_x^i$, $h_y^j$, and $h_{i,j}$ as the VGAE model's input feature vectors for the $i$-th constraint node, $j$-th variable node, and edge $(i, j)$, respectively. Similarly, $g_x^i$, $g_y^j$ and $g_{i,j}$ serve as the feature representations for these nodes and edges in the context of the discriminator model. They are formally defined as follows:

$$h_x^i = (A_i, \xi), h_x^i = (B_j, \xi), g_x^i = (A_i, d, u_i, \xi), g_y^j = (B_j, d, v_j, \xi),$$
$$h_{i,j} = g_{i,j} = (a_{ij}), \tag{3}$$

where $A_i$ and $B_j$ are one-hot vectors of length 3, which indicate the type of the constraint ($\leq, =, \geq$) and the type of the variable (integer, continuous, binary), respectively. $u_i$ and $v_j$ denote the right-

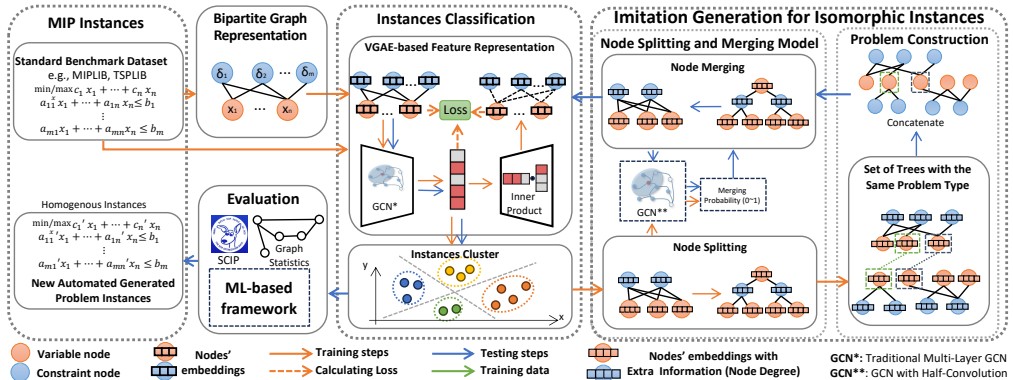

Figure 2: An overview of MIPGen. 1) The orange line signifies components that are active only during training. The process begins with obtaining the dataset that needs to be classified. MIPGen converts MIP problems into bipartite graph representations, utilizing the Gaussian Mixture Model for clustering. For problems of the same type, MIPGen employs the node splitting method to transform them into tree structures. This step also includes collecting training data for the discriminator model. In the final step, MIPGen trains the discriminator model and saves the tree structures to a template set for future use. 2) The blue line represents components that are active only during testing. Initially, MIPGen concatenates different tree structures to build a more comprehensive problem template. It then uses the discriminator model iteratively to merge nodes, leading to a complex bipartite graph structure. In the concluding phase, MIPGen reverts the bipartite graphs back to MIP problems and evaluates them.

hand side (RHS) of the $i$-th constraint and the coefficient of the $j$-th decision variable in the objective function, respectively. $d$ captures the node degree within the bipartite graph. Lastly, $\xi \sim U(0, 1)$ represents a random number following a uniform distribution between 0 and 1.

**Train VGAE to Learn Feature Representation.** With the BGR of MIP instances, VGAE is employed to generate a vector representation for each node within the bipartite graph, which captures essential structural and attribute-based information, facilitating the identification and categorization of MIP instances. To obtain a holistic feature representation for the entire graph, the individual node vectors are averaged to transform the intricate, discrete structure of a bipartite graph into a continuous, differentiable form suitable for machine learning models.

To be more specific, the encoder of VGAE in MIPGen comprises a three-layer GCN, tasked with mapping each node to a latent space. The decoder uses the inner product of the node vectors in this latent space to reconstruct the graph. Formally, the VGAE model is defined as follows.

$$\text{Encoder: } z = f_\theta(x, A) = \text{GCN}_{\text{3-layer}}(x, A), \tag{4}$$

$$\text{Decoder: } \hat{A} = \sigma(zz^T), \tag{5}$$

where $z$ is the latent vector representation for each node, $x$ denotes the input feature vectors (as described in the previous section), and $A$ is the adjacency matrix of the bipartite graph. $f_\theta$ represents the parameterized function of the encoder with parameters $\theta$, and $\sigma$ is the sigmoid activation function. The objective function for training the VGAE combines the reconstruction loss and the KL-divergence, facilitating both accurate graph reconstruction and efficient latent space mapping.

**Expectation–Maximization Algorithm with Gaussian Mixture Model.** Given the graph feature vectors from the VGAE encoder, MIPGen uses the Expectation–Maximization (EM) Algorithm combined with a Gaussian Mixture Model (GMM) to distinguish various MIP problem types. The input consists of the feature vectors corresponding to each bipartite graph, which encapsulates the unique characteristics of the corresponding MIP instances. By clustering these feature vectors using the EM algorithm with a GMM, MIPGen effectively groups similar MIP problems together, thereby achieving a categorization of the entire dataset of MIP problem instances.

The EM algorithm iteratively optimizes the parameters of the Gaussian mixture distributions to maximize the likelihood of the observed feature vectors. Specifically, it consists of an Expectation

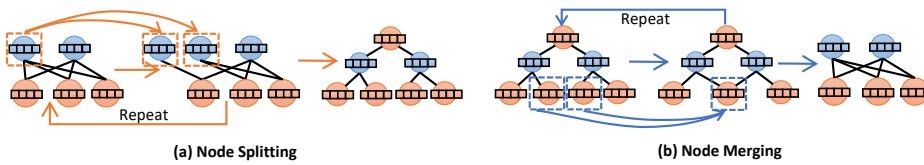

Figure 3: The Node Splitting and Node Merging process. Node Splitting converts a bipartite graph to tree structures and Node Merging converts tree structures to the bipartite graph.

(E) step and a Maximization (M) step, formalized as follows.

$$\text{E-step: } Q(\theta, \theta^{(t)}) = \sum_{i=1}^{N} \sum_{k=1}^{K} w_{ik} \log \frac{\pi_k \mathcal{N}(x_i | \mu_k, \Sigma_k)}{w_{ik}}$$

$$\text{M-step: } \theta^{(t+1)} = \arg\max_{\theta} Q(\theta, \theta^{(t)})$$

(6)

Here, $\theta = \{\pi_k, \mu_k, \Sigma_k\}_{k=1}^{K}$ are the parameters to be estimated, $w_{ik}$ is the posterior probability that observation $x_i$ belongs to cluster $k$, $\pi_k$ is the mixing coefficient, $\mu_k$ is the mean, and $\Sigma_k$ is the covariance matrix for the $k$-th Gaussian component. $N$ is the number of feature vectors, and $K$ is the number of clusters. Through iterative refinement, the algorithm converges to a local maximum of the likelihood function, providing a robust clustering solution.

### 3.2 NODE SPLITTING AND MERGING

In this stage, MIPGen focuses on learning the distribution over the BGR of MIP problems. Given the complex structure of bipartite graphs of MIPs, generating the entire graph in one step is unfeasible. Inspired by G2SAT (You et al., 2019), we adopt a step-by-step learning and generation approach. Specifically, we split the original bipartite graph into tree structures and employ the discriminator model to predict which pair of nodes should be merged to reconstruct the graph structure. Let $n$ be the number of nodes in the bipartite graph, and $m$ the number of edges. If MIPGen performs $x$ node splitting and merging operations to generate a new problem, the reconstruction percentage is defined as $\frac{x}{m-n}$.

**Node Splitting.** As shown in Algorithm 1, MIPGen first transforms the original bipartite graph into a tree-like structure, and then identifies a node $x$ in graph $G$ with the highest degree and splits it into two new nodes, $p$ and $q$. In particular, if node $x$ has a degree $d$ in $G$, the new node $p$ will inherit $d-1$ edges from $x$ (selected randomly) and node $q$ will inherit a single edge from $x$, which generates a new graph $G'$. For each graph $G_i$, let $x_i$ be the node to be split, Node Splitting generates a new graph $G_{i-1}$ and two new nodes $p_i$ and $q_i$. Then another node $r_i$ is selected from the same partition of $G$ where $p_i$ and $q_i$ belong to but is distinct from both. This process generates training data in the form of positive examples $(G_{i-1}, p_i, q_i)$ and negative examples $(G_{i-1}, p_i, r_i)$. The tuples $(G_{i-1}, p_i, q_i, r_i)$ are stored in a training dataset $D$ for subsequent usage.

MIPGen needs to maintain the node feature information mentioned in Section 3.1, $g_x^i = (A_i, d, u_i, \xi)$ and $g_y^j = (B_j, d, v_j, \xi)$. To achieve this goal, the one-hot vectors $A_i, B_j$ and the coefficients $u_i, v_j$, which represent the variables in the optimization objectives and constraint right-hand sides (RHS), remain the same as before splitting. In addition, the random feature $\xi$ is regenerated, and the degree $d$ is updated as $d_{p_i} = d_{x_i} - 1$ and $d_{q_i} = 1$ since $p_i$ inherits $d_{x_i} - 1$ edges from the split node $x_i$, and $q_i$ only gets one newly created edge.

**Discriminator Model Design.** The discriminator employs a half-convolution GCN optimized for bipartite graphs, augmented with a Multilayer Perceptron (MLP). This model begins with a three-layer semi-convolutional GCN that operates with identical parameters across layers. This GCN architecture is responsible for obtaining the node embeddings for each node in the bipartite graph. Then the model assesses the pair of nodes under consideration for merging. The embeddings of these two nodes are concatenated and fed into an MLP, which then outputs a value between 0 and 1—indicative of the confidence level for merging the nodes. For the training data mentioned earlier, the confidence score for positive examples should ideally be predicted as 1, whereas for the negative examples, it should be 0. The loss function used in the model is Binary Cross-Entropy Loss.

**Algorithm 1** Node Splitting Process

**Input:** Bipartite Graph $G$, Graph Template Set $\mathcal{T}$
**Output:** Graph Template Set $\mathcal{T}$
1: $n \leftarrow |\mathcal{E}_G| - |\mathcal{V}_G|, G_n \leftarrow G$
2: varcnt $\leftarrow 0$, concnt $\leftarrow 0$
3: **for** $i = n, n-1, \ldots, 1$ **do**
4:      $u \leftarrow \max\{degree_x | x \in \mathcal{V}_{G_i}\}$
5:      **if** $u$ is variable node **then**
6:          varcnt $\leftarrow$ varcnt $+ 1$
7:      **else**
8:          concnt $\leftarrow$ concnt $+ 1$
9:      **end if**
10:     $(u^+, v^+, G_{i-1}) \leftarrow \text{Split}(u, G_i)$
11:     Maintain vector of $u^+$ and $v^+$
12:     $v^- \sim \mathcal{V}_{G_{i-1}} - \{u^+, v^+\}$
13:     Train with $(u^+, v^+, v_-, G_{i-1})$.
14: **end for**
15: $\mathcal{T} \leftarrow \mathcal{T} \cup \{G_0, n, \text{varcnt}, \text{concnt}\}$

**Algorithm 2** Node Merging Process

**Input:** Graph Template Set $\mathcal{T}$, hyperparameter $K$, discriminator $D$
**Output:** Generated Bipartite Graph $G$
1: $\{G_0, n, \text{varcnt}, \text{concnt}\} \sim \mathcal{T}$
2: **for** $i = 0, 1, \ldots, n-1$ **do**
3:     $p \sim U(0, 1), \mathcal{S} \leftarrow \emptyset$
4:     **if** $p < \frac{\text{varcnt}}{\text{varcnt}+\text{concnt}}$ **then**
5:        **for** $j = 0, 1, \ldots, K-1$ **do**
6:           $(u, v) \sim \{(x, y) | x \in V_{G_i^{\text{var}}}, y \in V_{G_i^{\text{var}}} \backslash x\}$
7:           $\mathcal{S} \leftarrow \mathcal{S} \cup (u, v)$
8:        **end for**
9:     **else**
10:       **for** $j = 0, 1, \ldots, \mathcal{K}-1$ **do**
11:          $(u, v) \sim \{(x, y) | x \in V_{G_i^{\text{con}}}, y \in V_{G_i^{\text{con}}} \backslash x\}$
12:          $\mathcal{S} \leftarrow \mathcal{S} \cup (u, v)$
13:       **end for**
14:     **end if**
15:     $(u, v) \leftarrow \max\{\text{D}(G_i, x, y) | (x, y) \in \mathcal{S}\}$
16:     $G_{i+1} \leftarrow \text{NodeMerge}(G_i, u, v)$
17: **end for**

**Node Merging.** Node Merging in MIPGen transforms tree structures back into complex bipartite graphs, as shown in Algorithm 2. Specifically, consider an input graph $H_0$, through $n$ iterations of node merging operations, the algorithm yields $H_n$. For each graph $H_i$, MIPGen randomly samples $K$ pairs of nodes $\{(u_k, v_k)\}_{k=1}^K$ and employs the above discriminator model to parallelly compute the likelihood of merging each pair in the context of $H_i$. Then, it selects the node pair $(u_o, v_o)$ with the highest merging probability to form the subsequent graph $H_{i+1}$.

To maintain the bipartite nature of the resulting graph, it is essential that the nodes in each potential pair belong to the same half of the original bipartite graph. The selection probability for merging nodes from either the variable or constraint partition is determined based on their respective splitting frequencies $x$ and $y$ during the Node Splitting phase. Specifically, the probability of merging a decision-variable node is $\frac{x}{x+y}$, and that for a constraint-equation node is $\frac{y}{x+y}$.

Converse to the Node Splitting stage, for constraint nodes $g_x^i = (A_i, d, u_i, \xi)$ and decision-variable nodes $g_y^j = (B_j, d, v_j, \xi)$, the feature set $A_i$ or $B_j$ of the new merged node is randomly selected from one of the two nodes being merged. The degree of the new node is the sum of the degrees of the two original nodes. The coefficients $u_i$ and $v_j$ relating to optimization targets and constraint Right-Hand Sides (RHS) are averaged, and a new random feature $\xi$ is generated.

## 3.3 SCALABLE PROBLEM CONSTRUCTION

By utilizing Node Splitting and Merging, it becomes feasible to control the size of the MIP problem since the number of iteration rounds in the Node Merging process serves as a hyperparameter. However, the number of edges is limited, making it challenging to generate more complicated MIP problems. To overcome this limitation, MIPGen exploits the characteristics of trees to create larger-scale problems of the same category. In particular, we initially collect tree structures acquired through node splitting across multiple analogous problems. Then, the GCN component in the discriminator previously trained is employed to calculate node embeddings for each node. When combining two trees, nodes with the highest similarity are merged as a pair. This process is iterated, along with the node merging, to generate larger-scale problems of the same category.

**Node Embeddings Computation.** During the problem scaling phase, MIPGen initially utilizes the GCN component of the discriminator model for inference. This enables the computation of node embeddings for each node in the two trees marked for merging. However, due to computational constraints, it's impractical to compare the node embeddings of every node between the two trees.

To address this, a hyperparameter $E$ is introduced to select $E$ nodes from each tree for comparison. Importantly, the selected nodes must belong to the same side of the bipartite graph to ensure compatibility for merging. In addition, the classical cosine similarity metric is employed to compare the embeddings. With the computational complexity being $O(E^2)$, this algorithm conducts pairwise comparisons among the embeddings of all selected nodes. Subsequently, it identifies the pair with the highest similarity score as the optimal candidates for merging.

**Tree Merging.** The key idea of tree merging is to amalgamate multiple trees into a single and larger tree, each derived from the same type of MIP problem instances and subjected to node-splitting processes. This consolidated tree is then subjected to node merging to generate a more complex problem instance than those used during training. Given a set of trees denoted as $\mathcal{T}$, the process begins by randomly sampling a base tree $T_0$ from $\mathcal{T}$. Subsequent operations involve iteratively merging $T_{i-1}$ with a randomly selected tree $t$ from $\mathcal{T}$, resulting in a new tree $T_i$. Finally, $T_m$ becomes the enlarged tree after several iterations.

Merging two trees involves calculating node embeddings to identify the pair of nodes with the highest similarity for merging, similar to the node merging process. However, in addition to maintaining the node feature information of the newly merged nodes, similar to node merging, the process also keeps track of the number of node-splitting occurrences associated with the constraint equation nodes and decision variable nodes within the tree. The node-splitting metadata for each node in the newly formed tree is computed by summing the corresponding metadata from the original trees being merged. This ensures that the aggregated tree retains critical information from its constituent trees, facilitating more accurate and scalable problem-solving.

## 4 EXPERIMENTS

We evaluate the performance of MIPGen on three combinatorial optimization benchmark problems, including Maximum Independent Set (MIS) (Tarjan & Trojanowski, 1977), Combinatorial Auction (CA) (De Vries & Vohra, 2003), and Minimum Vertex Cover (MVC) (Dinur & Safra, 2005). To comprehensively validate the effectiveness of our proposed framework, we conduct the following two aspects of experiments. First, we study the performance of the generation of MIPGen with the same scale problem generation, assessing its ability to preserve MIP properties, its role as a data augmentation tool, and its impact on the optimization process (Sec. 4.1). Furthermore, we study the scalability of MIPGen to evaluate its remarkable imitation and generation capabilities (Sec. 4.2).

In the following experiments, we compare the generation results of MIPGen with two baselines. The first baseline, referred to as 'Bowly'(Bowly et al., 2020), is a heuristic MIP instance generator. It can generate feasible bounded MIP problems after specifying certain statistical feature distributions. In our experiments, we set its all controllable parameter to match the corresponding features of the MIP instances in our training dataset. The second baseline, named 'Random', is derived by randomizing the output of the discriminator in the Node Merging step of MIPGen, and the reconstruction percentage is set to 100%.

### 4.1 OVERALL PERFORMANCE OF GENERATION

To study the MIPGen performance of generation, we initially examine whether the generated MIP instances preserve the properties of the input training MIP instances through an analysis of solver performance (Sec. 4.1.1) and graph statistics (Sec. 4.1.2). Subsequently, we integrate MIPGen into the ML-based solver framework (Ye et al., 2023) to study its potential as a data augmentation technique (Section 4.1.3). This evaluation examines the quality of the generated MIP instances in predicting feasible solutions and their capacity to improve optimization results.

### 4.1.1 MIP SOLVER PERFORMANCE

To confirm that MIPGen can produce MIP instances that closely resemble the input MIP instances, we employ SCIP (Bestuzheva et al., 2021), a state-of-the-art MIP solver, to solve and compare the newly generated MIP problems with those from the training dataset. Solver performance results are presented in Table 1. The results highlight a high degree of similarity between the input MIP problems and the generated MIP problems. Our experimental findings indicate that the problems

Table 1: Comparison of the optimality gap obtained within a fixed wall time of 1200s between the generated MIP instances and the input training instance. The percentage represents how much of the graph structure is reconstructed using MIPGen, and the interval represents the result of 20 generated instances by specifying different random seeds. The percentage after MIPGen denoted reconstruction percentage.

| Optimality Gap | MIS | CA | MVC |
|---|---|---|---|
| Training Problem | 9.64% | 9.52% | 10.04% |
| MIPGen-25% | $7.06\% \sim 9.93\%$ | $9.30\% \sim 13.12\%$ | $7.06\% \sim 9.56\%$ |
| MIPGen-50% | $7.78\% \sim 11.31\%$ | $9.76\% \sim 12.69\%$ | $5.26\% \sim 7.81\%$ |
| MIPGen-100% | $4.64\% \sim 7.33\%$ | $6.82\% \sim 9.53\%$ | solved in $258.78s \sim 0.49\%$ |
| Random | solved in $1.29 \sim 150.05s$ | solved in $3.38 \sim 126.51s$ | solved in $1.31 \sim 35.42s$ |
| Bowly | solved in $0.14 \sim 0.41s$ | solved in $0.09 \sim 10.49s$ | solved in $0.13 \sim 2.44s$ |

Table 2: Comparison of the similarity metric between the generated MIP instances and the input training instance. Higher is better.

| | MIS | CA | MVC |
|---|---|---|---|
| MIPGen | **0.820** | **0.830** | **0.824** |
| Random | 0.578 | 0.559 | 0.572 |
| Bowly | 0.631 | 0.632 | 0.644 |

generated by MIPGen exhibit a level of solving complexity similar to that of the original problems, contrasting with the relatively trivial problems generated by the Random and Bowly methods. It is interesting to observe that as the reconstruction percentage increases, the solving difficulty of generated problems has a downward trend. For more experimental details, we refer to Appendix C.

### 4.1.2 GRAPH STATISTICS

To compare the similarity between the original Mixed Integer Programming (MIP) problems and the newly generated MIP problems, we designed a comprehensive evaluation metric for assessment. We compared the results generated by MIPGen with those produced by two baseline methods: Random and Bowly. The experimental results indicate that MIPGen can generate MIP problems highly similar to the original ones. For a detailed explanation of the evaluation metrics, please refer to the Appendix D.

### 4.1.3 DATA AUGMENTATION FOR ML-BASED FRAMEWORK

To validate MIPGen's efficiency in addressing the challenge of ML-based frameworks that rely on a substantial number of instances of isomorphic problems for training data, we make comparisons between only using the training dataset and using both the training dataset and the 20 newly generated MIP problems (use model trained by the input dataset) as training data for the ML-based framework. The evaluation focuses on the framework's solving performance for MIP problems and is shown in Table 3. The results indicate that the MIP problems newly generated by MIPGen enhance the predictive ability and solution effectiveness of the ML-based framework.

### 4.2 SCALABILITY OF MIPGEN

To further study MIPGen's capability to generate scalable MIP instances, we evaluate the large-scale problems derived through extrapolation from the small-scale training dataset. All the results are shown in Table 4. We can see that the solving performance, measured by SCIP, suggests that the generated scaling instances closely match those of the large-scale training dataset. This confirms the remarkable imitation and generation capabilities of MIPGen.

We also study the performance of MIPGen for data augmentation for generated scaled-up problems (MVC). The baseline method uses a directly constructed single problem as the training data of the ML-based framework. As a comparison, a single generated 8-fold enlarged problem is used as the training data. All the results are shown in Table 7. The results verify that generated scalable

Table 3: Comparison of the final optimized solution within a fixed time trained by generated MIP instances and the training dataset. MIPGen-25% means using MIPGen to reconstruct the 25% graph structure of the input problem. The scale-limited versions of SCIP which limit the variable proportion $\alpha$ is set to 30%, the time limit of the solver framework is 50s.

| | MIS | | CA | | MVC | |
|---|---|---|---|---|---|---|
| | Original | Augmented | Original | Augmented | Original | Augmented |
| MIPGen-25% | | 2122.68 ↑ | | 1460.78 ↑ | | 2870.89 ↑ |
| MIPGen-50% | 2092.99 | 2141.37 ↑ | 1452.95 | 1503.47 ↑ | 2918.80 | 2837.94 ↑ |
| MIPGen-100% | | 2080.6 | | 1444.40 | | 2911.74 ↑ |
| Bowly | | 1797.2 | | 1069.7 | | 3008.7 |

Table 4: Comparison of the optimality gap obtained within a fixed time between the scalable generated MIP problems and the training dataset.

| | Variable Num | Constraint Num | Edge Num | Gap |
|---|---|---|---|---|
| MIS-input | 2500 | 7500 | 15000 | 8.09% |
| MIS-1x | 2500 | 7500 | 15000 | 3.30% |
| MIS-2x | 5000 | 15000 | 29992 | 4.41% |
| MIS-4x | 9997 | 30000 | 59999 | 24.86% |
| MIS-8x | 19994 | 60000 | 120000 | 25.89% |
| CA-input | 2500 | 5000 | 15000 | 9.81% |
| CA-1x | 2500 | 50000 | 14999 | 6.25% |
| CA-2x | 4999 | 10000 | 29996 | 8.70% |
| CA-4x | 9997 | 20000 | 59993 | 31.83% |
| CA-8x | 19993 | 40000 | 119993 | 233.59% |
| MVC-input | 2500 | 7500 | 15000 | 5.58% |
| MVC-1x | 2500 | 7500 | 14997 | 3.20% |
| MVC-2x | 4999 | 15000 | 29998 | 5.68% |
| MVC-4x | 9998 | 30000 | 59998 | 22.48% |
| MVC-8x | 19993 | 60000 | 119999 | 22.54% |

instances also greatly imitate input training instances. Also, the generated instances used for data augmentation confirm that MIPGen helps with the data shortage of ML-based solving frameworks.

Table 5: Comparison of the final optimized solution within a fixed time trained by generated MIP instances and the training dataset. $S - 30\%$ means the scale-limited versions of SCIP which limit the variable proportion $\alpha$ is set to 30%, the time limit of the solver framework is 50s.

| | S-20% | | S-30% | | S-50% | |
|---|---|---|---|---|---|---|
| | Baseline | Augmented | Baseline | Augmented | Baseline | Augmented |
| MVC | 2943.38 | 2886.42 ↑ | 2990.55 | 2854.87 ↑ | 2885.08 | 2767.58 ↑ |

## 5 CONCLUSION

This study presents MIPGen, a pioneering deep generative model tailored for MIP problems. Leveraging advanced techniques like a random-feat policy for bipartite graph representation, an EM clustering algorithm with VGAE, and node operations in an expanded feature space, MIPGen effectively addresses the shortcomings of prior models, such as limited representation in large-scale MIP problem generation, inefficient instance classification, and oversimplified problem structures. Empirical results on standard MIP instances confirm MIPGen's ability to learn problem features and create high-quality isomorphic instances, marking a significant advancement in machine learning-based frameworks for MIP problems. Currently focused on single-objective, linear, and static problems, future work will extend to more complex scenarios including multi-objective, nonlinear, and dynamic problems.

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

# A EXPERIMENTAL DETAILS

## A.1 EXPERIMENTAL SETTINGS

For MIP problem generation, we conduct experiments on a system equipped with two AMD EPYC 7742 CPUs clocked at 1.50GHz and six NVIDIA TESLA A800 GPUs. The details on decision variables and constraint scales for three NP-hard benchmark MIP instances can be found in Table 6. PySCIPOpt 4.3.0 (Maher et al., 2016) was employed as the interface to invoke SCIP.

Experiments related to data augmentation were executed on a system with two Intel Xeon Platinum 8375C CPUs operating at 2.90GHz and four NVIDIA TESLA V100(32G) GPUs. We utilized a GBDT with parameters set to contain 30 decision trees, and each was limited to a maximum depth of 5.

Table 6: The size of three MIP benchmark problem. MIS stands for Maximum Independent Set problem, CA stands for Conbinatorial Auction problem, and MVC stands for Minimum Vertex Cover problem.

|      | Variable Number | Constraint Number | NNZ number |
|------|-----------------|-------------------|------------|
| MIS  | 10000           | 30000             | 60000      |
| CA   | 10000           | 20000             | 60000      |
| MVC  | 10000           | 30000             | 60000      |

## A.2 DATASETS

We employed data generators to produce training and test datasets for three popular benchmark MIP problems:

- Maximum Independent Set (MIS) / Minimum Vertex Covering (MVC): For problems with $n$ decision variables and $m$ constraints, a random graph with $n$ nodes and $m$ edges was generated. This translates to an MIP problem fitting the specified scale criteria.
- Combinatorial Auction (CA): For problems with $n$ decision variables and $m$ constraints, a random problem was generated comprising $n$ items and $m$ bids. Notably, each bid encompasses three items.

To generate the training solution dataset required by the solver framework (Ye et al., 2023) for data augmentation assessment, we utilized PySCIPOpt 4.3.0, allowing it to run for 2400s to get an approximate optimal solution.

# B INSTANCES CLASSIFICATION EXPERIMENTS

We designed two experiments to validate the effectiveness of our classification method. These experiments were conducted on problems from the MIS, CA, and MVC problem mentioned in the paper, as well as on MIP problems from the MIPLib benchmark (Gleixner et al., 2021).

For the MIS, CA, and MVC problems, we selected 20 directly constructed problems for each type. Furthermore, using MIPGen, we generated additional problems by restructuring at 25

In the MIPLib classification experiment, we trained the VGAE on 240 problems from the MIPLib Benchmark, setting the clustering count to 100. MIPLib problems often have detailed descriptions and contributor information on their web pages, shedding light on the real-world context and origins of each problem. We focused on problems where the descriptions and contributors matched, to assess if they were clustered into the same category. Such problems, sharing descriptions and contributors, can be regarded as belonging to the same type. Within a ML-based framework, these can serve as training data for specific problem types. A portion of the successfully clustered problems are listed in Table 7. Additionally, Figure 4 depicts the clustering intermediate results of these 240 problems. The VGAE's encoder outputs a 16-dimensional vector for each MIP problem, which serves as the basis for clustering. After reducing the dimensionality to 3 using the PCA method, we

plotted a point cloud diagram of the MIP problem representations. In this visualization, we excluded 11 outliers from the total of 240 points.

This experiment showcases the capability of our method in effectively classifying problems. It confirms that, given a complex dataset comprising various types of MIP instances, the MIPGen framework can categorize them into distinct groups, learn the corresponding model, and generate new instances for each problem category.

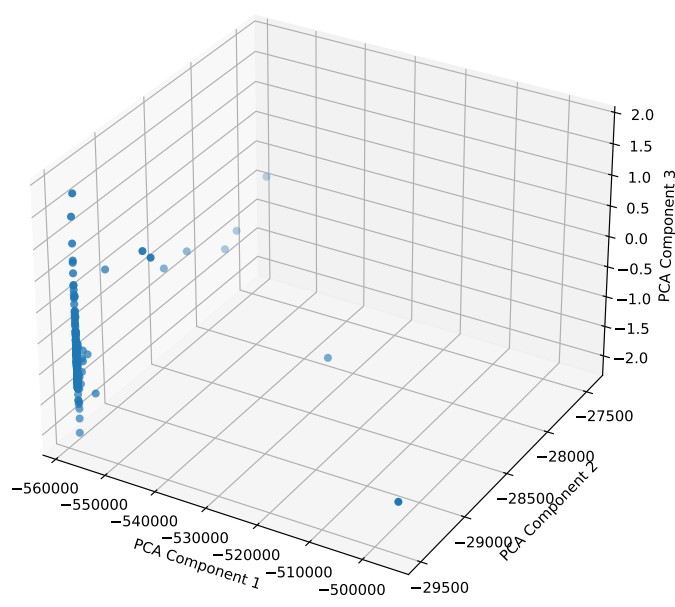

Figure 4: 3D PCA of MIPLIB Benchmark Instances (Outliers Removed)

## C   DETAILED MIP SOLVER PERFORMANCE

we present the solving difficulty information for the MIP problems generated by various methods. Specifically, we provide the solving difficulty for five instances each generated by MIPGen-25%, MIPGen-50%, MIPGen-100%, Random, and Bowly methods. Additionally, we include a distribution graph of the solving difficulty for problems generated by MIPGen at different reconstruction percentages.

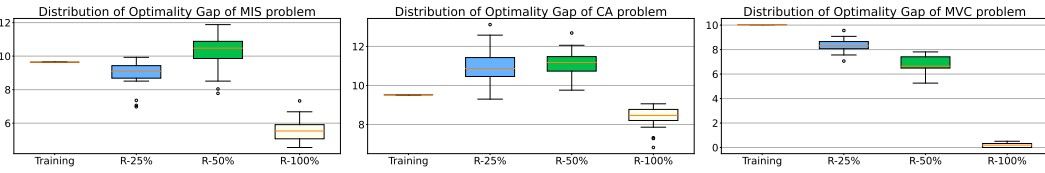

Figure 5: Distribution of Optimality Gap of Generated Instances

Table 7: Partial Classification Result of MIPGen

| Problems | Description | Cluster ID |
|---|---|---|
| app1-1, app1-2 | The archive contains 5 instances coming from 3 applications.app1 is interesting because the continuous variables (w) drive the model. | 11 |
| blp-ar98, blp-ic98 | Railway line planning instance. | 49 |
| bab2, bab6 | Vehicle Routing with profits and an integrated crew scheduling formulated by two coupled multi-commodity flow problems. | 49 |
| bnatt400, bnatt500 | Model to identify a singleton attractor in a Boolean network, applications in computational systems biology. | 11 |
| chromaticindex1024-7, chromaticindex512-7 | Simple edge-coloring model on chains of Petersen-like subgraphs, designed to fool MIP solvers into producing very large Branch-and-Bound trees. | 73 |
| comp07-2idx comp21-2idx | Instances comp01-21 of curriculum based course timetabling from the International Timetabling Competition 2007. | 11 |
| cryptanalysiskb128n5obj16, cryptanalysiskb128n5obj14 | Linearized Constraint Programming models of the MiniZinc Challenges 2012-2016. | 73 |
| csched008, csched007 | Cumulative scheduling problem instance | 58. |
| dano3_5, dano3_3 | Telecommunications applications. | 43 |
| ex10, ex9 | Formulations of Boolean SAT instance. | 95 |
| leo1, leo2 | Instance coming from the CORL test set with unknown origin | 78 |
| proteindesign121hz512p9, proteindesign122trx11p8 | Linearized Constraint Programming models of the MiniZinc Challenges 2012-2016. | 58 |
| radiationm18-12-05, radiationm40-10-02 | Linearized Constraint Programming models of the MiniZinc Challenges 2012-2016. | 75 |
| rmatr100-p10, rmatr200-p5 | Instance coming from a formulation of the p-Median problem using square cost matrices | 73 |
| rocI-4-11, rocII-5-11 | Optimal control model in the deterministic dynamic system given by bounded-confidence dynamics in a system of opinions | 11 |
| rococoB10-011000, rococoC10-001000 | Model for dimensioning the arc capacities in a telecommunication network. | 90 |
| sp97ar, sp98ar | Railway line planning instance. | 78 |
| square41, square47 | Squaring the square For a given integer n, determine the minimum number of squares in a tiling of an n×n square using only integer-sided squares of smaller size. | 51 |
| swath1, swath3 | Model arising from the defense industry, involves planning missions for radar surveillance. | 75 |

Table 8: Detailed solving difficulty information.

| | No. | MIS Problem | CA Problem | MVC Problem |
|---|---|---|---|---|
| Original Instance | - | 9.64%@1200s | 9.52%@1200s | 10.04%@1200s |
| MIPGen | Instance 1 | 5.78%@1200s | 7.32%@1200s | 0.33%@1200s |
| | Instance 2 | 7.33%@1200s | 8.96%@1200s | Solved in 674.13s |
| | Instance 3 | 4.84%@1200s | 9.06%@1200s | Solved in 658.47s |
| | Instance 4 | 5.51%@1200s | 6.82%@1200s | 0.21%@1200s |
| | Instance 5 | 5.39%@1200s | 8.78%@1200s | Solved in 258.78s |
| Random | Instance 1 | Solved in 1.29s | Solved in 128.67s | Solved in 1.38s |
| | Instance 2 | Solved in 14.71s | Solved in 121.33s | Solved in 6.81s |
| | Instance 3 | Solved in 1.34s | Solved in 64.04s | Solved in 1.34s |
| | Instance 4 | Solved in 1.69s | Solved in 6.59s | Solved in 1.79s |
| | Instance 5 | Solved in 3.52s | Solved in 8.10s | Solved in 4.39s |
| Bowly | Instance 1 | Solved in 0.15s | Solved in 0.21s | Solved in 0.25s |
| | Instance 2 | Solved in 0.26s | Solved in 0.12s | Solved in 0.13s |
| | Instance 3 | Solved in 0.19s | Solved in 10.49s | Solved in 0.27s |
| | Instance 4 | Solved in 0.28s | Solved in 0.23s | Solved in 0.19s |
| | Instance 5 | Solved in 0.41s | Solved in 0.27s | Solved in 0.24s |

Table 9: Evaluation metrics used in similarity comparison.

| Name | Explanation |
|---|---|
| coef_dens | Fraction of non-zero entries in coefficient matrix. |
| cons_degree_mean | Mean degree of constraint vertices. |
| cons_degree_std | Std of degrees of constraint vertices. |
| var_degree_mean | Mean degree of variable vertices. |
| var_degree_std | Std of degrees of variance vertices. |
| lhs_mean | Mean of non-zero entries in coefficient matrix. |
| lhs_std | Std of non-zero entries in coefficient matrix. |
| rhs_mean | Mean of RHS values. |
| rhs_std | Std of RHS values. |
| clustering_coef | Clustering coefficient of the graph. |
| modularity | Modularity of the graph. |

## D    EXPERIMENTAL SETTINGS FOR SIMILARITY METRIC

Table 9 lists 11 metrics used for assessing features in MIP instances. We conducted a similarity comparison between problems generated by MIPGen at various reconstruction percentages and two baselines, Random and Bowly, against MIP problems in the training set. For each individual metric, we calculated the JS (Jensen-Shannon) divergence between them. Let $JS_i$ denotes the JS divergence for the $i^{th}$ metric. The similarity score for the $i^{th}$ metric is defined as follows:

$$\text{score}_i = (\max(JS) - JS_i)/(\max(JS) - \min(JS)) \tag{7}$$

The overall similarity score is the average of the scores for all metrics:

$$\text{score} = \frac{1}{11} \sum_{i=1}^{11} \text{score}_i \tag{8}$$

