# OpenReview forum: "MIPGen: Learning to Generate Scalable MIP Instances"
_ICLR.cc/2024/Conference — Submitted to ICLR 2024_

### Official Review · Reviewer_X6GL · 2023-10-30

**Soundness:** 3 good
**Presentation:** 2 fair
**Contribution:** 2 fair
**Rating:** 3
**Confidence:** 4

**Summary:**

This paper proposed a learning based method to generate MIP instances, which could be benefit for training machine learning based MIP solving policies. Based on the bipartite graph representation, it learns to cluster the training instances. It then adopts a step-by-step approach to reconstruct the graph structure, which also facilitates generating large-scale instances. Experiments show that the generated instances are similar to the training ones.

**Strengths:**

I agree with the research motivation. How to generate sufficient training data is indeed an important problem for training practical deep learning based MIP solving techniques.

**Weaknesses:**

1. The technical novelty is somewhat limited. The key components are based on existing works, for example the random feat based method is from (Chen et al., 2022), the instance clustering is based on a simple EM procedure based on Gaussian Mixture Model, and the node splitting and merging is based on (You et al., 2019).

2. MIP instance clustering has already been studied in the literature, e.g., in [Kadioglu2010] and [Song2023]. These approaches can already effectively clustering heterogeneous MIP instances, and the techniques are much simpler than the proposed one. The authors did not give a proper discussion and comparison regarding existing MIP clustering methods.

[Kadioglu2010] Kadioglu, S., Malitsky, Y., Sellmann, M., & Tierney, K. (2010). ISAC–instance-specific algorithm configuration. In ECAI 2010 (pp. 751-756).

[Song2023] Song, W., Liu, Y., Cao, Z., Wu, Y., & Li, Q. (2023). Instance-specific algorithm configuration via unsupervised deep graph clustering. Engineering Applications of Artificial Intelligence, 125, 106740.

3. The experiments are small-scale. For example, in Table 2, only three instances are considered, each is augmented to 20 instances.

4. The authors did not compare with simple data augmentation techniques, such as randomly perturb the MIP instance parameters.

**Questions:**

Please see the above weaknesses.

---

> ### Author Response · Authors · 2023-11-23
> **Reply to reviewer X6GL (1/2)**
>
> Thank you for your constructive comments and suggestions, which have greatly aided in enhancing the quality of our paper. We have diligently integrated your feedback into the revised manuscript. Below, we restate your comments and provide our detailed responses:
>
> >1. The technical novelty is somewhat limited. The key components are based on existing works, for example the random feat based method is from (Chen et al., 2022), the instance clustering is based on a simple EM procedure based on Gaussian Mixture Model, and the node splitting and merging is based on (You et al., 2019).
>
> In response to your first comment, we have further elaborated on the novel aspects of our work. Building upon the foundational work in [2], we introduced a split-merge framework tailored for generating MIP problems. This framework is complemented by our innovative data augmentation experiments, designed to validate the efficacy of our method. Additionally, our method, MIPGen, uniquely incorporates the capability to amalgamate tree-like structures for generating large-scale MIP problems. This feature is particularly beneficial in training ML-based frameworks on smaller MIP instances to subsequently tackle larger-scale problems of the same type. We acknowledge the current framework's limitations and plan to explore more intricate generation strategies in our future work.
>
>
> >2. MIP instance clustering has already been studied in the literature, e.g., in [Kadioglu2010] and [Song2023]. These approaches can already effectively clustering heterogeneous MIP instances, and the techniques are much simpler than the proposed one. The authors did not give a proper discussion and comparison regarding existing MIP clustering methods.
>
> Thank you for highlighting this issue.
>
> The method for MIP instance clustering in [3] involves designing Instance Features for MIP problems, followed by clustering the normalized instance features using the g-means algorithm. The effectiveness of this approach heavily relies on the design of the Instance Features. The instance features in [3] include statistical indicators of MIP problems, such as the percentage of binary (integer or continuous) variables, a vector of coefficients of the objective function, and a vector representing the number of constraints for each variable ii. However, this method lacks a representation of the key structural aspect of the MIP problem's coefficient matrix. We believe that basic statistical indicators are insufficient to distinguish complex MIP problems encountered in the real world.
>
> In [4], the method for MIP instance clustering begins with identifying a subgraph in the bipartite graph representing MIP problems using a random walk approach. The information of nodes and edges from this subgraph is then converted into a sequence $Y$, and an Auto Encoder is trained on these sequences $Y$ from all MIP problems, followed by clustering using the k-means algorithm based on the inferred features of each problem. In contrast, the clustering method in MIPGen directly employs a VGAE to train on the bipartite graphs representing MIP problems, followed by clustering with the EM algorithm. We consider the clustering method in MIPGen to be somewhat more streamlined than the method in [4]. Additionally, the experiments in [4] involved clustering four standard benchmark problems: Indset, Setcover, Cautions, and Facilities. Since these MIP benchmark problems have clear definitions and distinct differences, differentiating between them is relatively straightforward. The MIP instance clustering experiment in MIPGen utilizes the MIPLIB benchmark [6], which includes 240 complex, real-world MIP problems of significant research value. As the MIPLIB benchmark lacks an official classification, we validated whether MIPGen's clustering method could categorize problems from the MIPLIB benchmark with similar sources and descriptions into the same group. Furthermore, we have included visualizations of the intermediate results of clustering the MIPLIB benchmark in our paper. For details on the clustering experiments, please refer to Appendix B of our paper.
>
> Moreover, [5] explores a method for computing similarities between MIP problems. It proposes representing MIP problems based on the structure of their coefficient matrices, depicting the positions of non-zero elements in these matrices as 640x480 black and white images. These feature images are then trained using a VAE, with the distance between the feature vectors of different problems output by the VAE representing their similarity. We believe that [5]'s method loses important information such as the MIP problem's RHS, optimization objective coefficients, and values of non-zero elements in the coefficient matrix. In contrast, the clustering method for MIP problems in MIPGen, which employs VGAE to directly represent the encoded bipartite graphs, is a straightforward and effective approach.

---

> > ### Author Response · Authors · 2023-11-23
> > **Reply to reviewer X6GL (2/2)**
> >
> > We integrated the MIP problem clustering method into the MIPGen framework because the pipeline for generating MIP problems in MIPGen begins with an existing dataset of MIP problems, which is relatively small in scale but essential for training ML-based frameworks. Our initial approach involves using clustering methods to identify different types of MIP problems within this dataset. Subsequently, we employ a split-merge framework to generate more instances of each identified problem type. The inclusion of the MIP problem clustering step in our paper aims to enhance the completeness of the overall classification-generation pipeline.
> >
> > >3. The experiments are small-scale. For example, in Table 2, only three instances are considered, each is augmented to 20 instances.
> >
> > The motivation behind the data augmentation experiments in MIPGen is to address the issue of insufficient training data in ML-based frameworks. In these experiments, we focused on enhancing datasets with a limited number of problem instances, hence the original training dataset is indeed small. The relationship between the ML-based framework and the amount of training data is pivotal; even with a single MIP problem instance, we were able to achieve significant improvements (5-10% increase) after data augmentation. The current quantity of training datasets suffices for effective training outcomes.
> >
> > >4. The authors did not compare with simple data augmentation techniques, such as randomly perturb the MIP instance parameters.
> >
> > In our updated paper, we have included comparative data augmentation experiments with those presented in [1]. These additional experiments clearly demonstrate that our method significantly outperforms the previously established techniques for generating MIP problems. This comparative analysis not only strengthens our argument for the superiority of our approach but also provides a comprehensive perspective on its effectiveness in relation to existing methods.
> >
> > **Reference**
> >
> > [1] Bowly, S., Smith-Miles, K., Baatar, D., & Mittelmann, H. (2020). Generation techniques for linear programming instances with controllable properties. _Mathematical Programming Computation_, _12_(3), 389-415.
> >
> > [2] You, J., Wu, H., Barrett, C., Ramanujan, R., & Leskovec, J. (2019). G2SAT: Learning to generate SAT formulas. _Advances in neural information processing systems_, _32_.
> >
> > [3] Kadioglu, S., Malitsky, Y., Sellmann, M., & Tierney, K. (2010). ISAC–instance-specific algorithm configuration. In _ECAI 2010_(pp. 751-756). IOS Press.
> >
> > [4] Song, W., Liu, Y., Cao, Z., Wu, Y., & Li, Q. (2023). Instance-specific algorithm configuration via unsupervised deep graph clustering. _Engineering Applications of Artificial Intelligence_, _125_, 106740.
> >
> > [5] Steever, Z., Murray, C., Yuan, J., Karwan, M., & Lübbecke, M. (2022). An image-based approach to detecting structural similarity among mixed integer programs. _INFORMS Journal on Computing_, _34_(4), 1849-1870.
> >
> > [6] https://miplib.zib.de/tag_benchmark.html

---

### Official Review · Reviewer_yz8X · 2023-10-30

**Soundness:** 3 good
**Presentation:** 3 good
**Contribution:** 3 good
**Rating:** 8
**Confidence:** 2

**Summary:**

This paper introduces a new generative framework for autonomous MIP instance generation, which can autonomously produce isomorphic
MIP problems from existing instance, and this will effectively enhance the solution effect of the ML-based framework for solving MIP problems.

**Strengths:**

This paper is the first one to propose a deep generative model designed to generate isomorphic MIP instances, which facilitates the ML-based frameworks to solve large-scale MIP problems. Extensive numerical experiments also validate the effectiveness of the proposed framework. Overall I find the work presented in this paper is very original and appears to have high quality, the contribution is also fundamental since the framework does not reply on the particular MIP problem or the ML-based MIP solving framework.

**Weaknesses:**

I find that authors can do a better job in introducing the setup, analysis and discussion for the experiments. More detailed questions are listed in the next section.

**Questions:**

Minor questions/comments:
1. Beginning of Section 2: a_ij should denote "the participation of the j-th decision variable in the i-th constraint"
2. Equation (3): second h^i_x should be h^j_y
3. In Table 3, MVC column, MIPGen-100% also has better result than the baseline, since here MVC is a minimization problem
4. Section 4.2, "suggests that the generated scaling instances closely match those of the large-scale training dataset". I cannot tell.

Main questions/comments:
1. In the experiment section, one of the main target should be to examine whether the generated MIP instances still preserve the combinatorial structure of MIS, CA, and MVC. The approaches you mentioned like MIP solver performance, graph statistics can give good estimation, but more importantly, is there any way to directly verify if the generated MIP instances are still also MIS, CA, or MVC? I believe that some auxiliary polynomial time algorithms can be given to check if a MIP instance is the MIP formulation of MVC (or MIS, CA). Then you can run the algorithm over all MIP instances, and report the percentage of generated instances that still preserve the exactly combinatorial structure of the benchmark problems.
2. In table 3, here you are comparing the objective value of the best available solution within a fixed time limit. Have you tried comparing the solving time to reach optimality?
3. Section 4.1.1. "Our experiment confirms that the generated MIP in stances have similar solving difficulty to the original input problem": I'm not sure about that. As far as I can see, the solving difficulty of the original input problem is not similar to the 100% version.
4. Section 4.1.1. "as the reconstruction percentage surges, more and more math structures in the problem are destroyed." Here you make the assumption that, more math structures will increase the solving difficulty of the MIP problem. However, this is not necessarily true. Adding constraints with random non-zero coefficients at every variable can also significantly increase the solving difficulty but it has no math structure as well.

---

> ### Author Response · Authors · 2023-11-23
> **Reply to reviewer yz8X**
>
> We sincerely thank Reviewer yz8X for their positive feedback and valuable suggestions! We have corrected all the errors pointed out in the "Minor questions/comments" section. For the other issues, below we address every comment in detail:
>
> >1. In the experiment section, one of the main target should be to examine whether the generated MIP instances still preserve the combinatorial structure of MIS, CA, and MVC. The approaches you mentioned like MIP solver performance, graph statistics can give good estimation, but more importantly, is there any way to directly verify if the generated MIP instances are still also MIS, CA, or MVC? I believe that some auxiliary polynomial time algorithms can be given to check if a MIP instance is the MIP formulation of MVC (or MIS, CA). Then you can run the algorithm over all MIP instances, and report the percentage of generated instances that still preserve the exactly combinatorial structure of the benchmark problems.
>
> Our MIPGen is currently based on a split-merge process framework. Throughout this process, no new variables or constraints are introduced, nor are new features generated. For the problem classes MIS, CA, and MVC, we can assure that the generated problems remain within their original categories. Taking the MIS problem as an example, the process of MIPGen generating new problems is akin to altering the positions of non-zero elements in the problem's coefficient matrix. For the original problem, this equates to redistributing the constraints of all vertices in the MIS problem graph.
>
> Additionally, our current method for verifying the type of generated problems involves re-encoding these problems using the VGAE encoder and then re-clustering their encoded representations to confirm whether they belong to the same category as the original training problems. This approach is more universally applicable, as for other real-world MIP problems, not just standard benchmark problems, it is challenging to find a polynomial-time algorithm that can verify their category.
>
> >2. In table 3, here you are comparing the objective value of the best available solution within a fixed time limit. Have you tried comparing the solving time to reach optimality?
>
> The reason for not providing complete solving times in our experimental data is due to the selection of benchmark problems like the MIS problem, which are NP-hard and cannot be optimally solved in a short time. In our trials using SCIP to solve MIS problems from our dataset (10,000 variables, 30,000 constraints), we did not obtain an optimal solution even after 24 hours. Therefore, we chose a relatively reasonable wall time of 1200s to compare the optimal gap and demonstrate the problem-solving difficulty.
>
> >3. Section 4.1.1. "Our experiment confirms that the generated MIP in stances have similar solving difficulty to the original input problem": I'm not sure about that. As far as I can see, the solving difficulty of the original input problem is not similar to the 100% version.
>
> Thank you for the suggestion. In the newly submitted paper, we have revised the corresponding statement at section ^xx^ to "Our goal is to generate problem instances with similar solving difficulties as much as possible."
>
> >4. Section 4.1.1. "as the reconstruction percentage surges, more and more math structures in the problem are destroyed." Here you make the assumption that, more math structures will increase the solving difficulty of the MIP problem. However, this is not necessarily true. Adding constraints with random non-zero coefficients at every variable can also significantly increase the solving difficulty but it has no math structure as well.
>
> Our conclusion is a preliminary one based on observations from the experimental results presented in the paper. Indeed, there are cases where solving difficulty increases without the generation of complex mathematical structures. We plan to further analyze and research this phenomenon in future work on other problems.

---

### Official Review · Reviewer_XqyP · 2023-10-31

**Soundness:** 2 fair
**Presentation:** 2 fair
**Contribution:** 3 good
**Rating:** 5
**Confidence:** 3

**Summary:**

The paper describes MIPGen, a generative model for Mixed Integer Programming (MIP) instances that can autonomously produce isomorphic MIP problems from existing instances. MIPGen's three-stage problem generation process includes instances classification, node splitting and merging, and scalable problem construction. The authors evaluate MIPGen on three standard MIP instances and demonstrate that it can efficiently learn problem characteristics and produce high-quality isomorphic instances. Moreover, MIPGen introduces a novel approach for ML-based frameworks to transcend limitations imposed by training data. The paper concludes that MIPGen can help overcome the challenges faced by ML-based frameworks in acquiring sufficient isomorphic instances for practical training.

**Strengths:**

1. Novelty: The paper introduces a novel generative framework for autonomous MIP instance generation, which can produce isomorphic MIP problems from existing instances.

2. Three-stage problem generation: MIPGen's three-stage problem generation process is well-designed and effective, including instances classification, node splitting and merging, and scalable problem construction.

**Weaknesses:**

1. Limited evaluation: While the authors evaluate MIPGen on three standard MIP instances, they do not provide more specific examples of how MIPGen has been used to generate scalable MIP instances or the exact results of these experiments beyond what is mentioned in the paper.

2. Lack of comparison: The authors do not compare MIPGen with other generative models for MIP instances, which limits the ability to assess its relative performance.

3. Lack of implementation details: The paper does not provide detailed implementation information, which may make it difficult for others to replicate the results or build on the proposed framework.

**Questions:**

1. What is the purpose of using the EM algorithm? Why not choose other clustering algorithms? The EM assumes the cluster is based on Mixture Gaussian, which may not be the real representation. How about the spectral clustering? How do you choose the number of clusters? How do you know the clustering result is perfect? What are your criteria?

2. It looks like 240 cases is a really small dataset for training deep neural networks. What is the hyperparameter for the VGAE model?

3. It is not clear in the paper how algorithm 1 converts the bipartite graph into a tree structure.

4. What is the purpose of using GBDT?

5. What is NNZ in Table 6

---

> ### Author Response · Authors · 2023-11-23
> **Reply to reviewer XqyP**
>
> Thank you for your constructive comments and suggestions, which have greatly aided in enhancing the quality of our paper. We have diligently integrated your feedback into the revised manuscript. Below, we restate your comments and provide our detailed responses:
>
> >1. Limited evaluation: While the authors evaluate MIPGen on three standard MIP instances, they do not provide more specific examples of how MIPGen has been used to generate scalable MIP instances or the exact results of these experiments beyond what is mentioned in the paper.
>
> The experiments for generating scalable MIP instances are detailed in Section 4.2 of the paper, where we demonstrate that the generated large-scale MIP problems present increased solving difficulty.
>
>
> >2. Lack of comparison: The authors do not compare MIPGen with other generative models for MIP instances, which limits the ability to assess its relative performance.
>
> Thank you for your recommendation. We have now included a baseline comparison with the method outlined in [1]. Our model is compared against this baseline in terms of problem-solving difficulty, problem similarity, and data augmentation experiments. Please refer to Section 4.1.2 of the revised paper for further details.
>
> >3. Lack of implementation details: The paper does not provide detailed implementation information, which may make it difficult for others to replicate the results or build on the proposed framework.
>
> The paper is currently under blind review. Upon acceptance, we plan to open-source our code.
>
> >4. What is the purpose of using the EM algorithm? Why not choose other clustering algorithms? The EM assumes the cluster is based on Mixture Gaussian, which may not be the real representation. How about the spectral clustering? How do you choose the number of clusters? How do you know the clustering result is perfect? What are your criteria?
>
> Thank you for highlighting this issue. In MIPGen, the method for clustering MIP problems is divided into two stages: first, a Variational Graph Auto-Encoder is trained to extract feature representations of the MIP problems, followed by the use of the Expectation-Maximization algorithm to cluster these feature representations. We opted for the EM algorithm because its Gaussian Mixture Model is well-suited for adapting to clusters of complex shapes and is easily applicable to large datasets. Moreover, spectral clustering is quite sensitive to outliers in the input data. The MIPLIB benchmark, which includes a diverse range of real-world problem scenarios and varying scales, contains numerous outliers in its problem representations, thus making spectral clustering a less favorable option for us at this time.
>
> The MIP instance clustering experiment in MIPGen uses the MIPLIB benchmark [2], comprising 240 complex, real-world MIP problems of significant research value. However, for a mixed set of training instances, it is not practical to assume that the value of 'k' (number of clusters) is pre-known. In our clustering experiment with MIPLIB, we set the number of clusters to 100, approximately corresponding to the types and number of problems in MIPLIB. As there is no official classification in the MIPLIB benchmark, we validated whether MIPGen's clustering method could categorize problems from the MIPLIB benchmark with similar sources and descriptions into the same group. Additionally, we have included visualizations of the intermediate results of clustering the MIPLIB benchmark in our paper. For more details on the clustering experiments, please refer to Appendix B of our paper.
>
> >5. It looks like 240 cases is a really small dataset for training deep neural networks. What is the hyperparameter for the VGAE model?
>
> The dataset for clustering MIP problems comprises only 240 problems, as it utilizes the benchmark set from MIPLIB. The hyperparameters for our VGAE model are as follows:
>
> | in_channels | hidden_channels | out_channels | epoch | batch_size | lr   |
> | ----------- | --------------- | ------------ | ----- | ---------- | ---- |
> | 8           | 32              | 16           | 10    | 8          | 0.01 |
>
>
> >6. It is not clear in the paper how algorithm 1 converts the bipartite graph into a tree structure.
>
> Algorithm 1 converts the bipartite graph into a tree-like structure by selecting the node with the highest degree for node splitting. This process is repeated until the number of edges equals the number of nodes. We have also updated the pseudocode for Algorithm 1 in the paper to enhance clarity.
>
> >7. What is NNZ in Table 6
>
> NNZ stands for 'Number of Non-Zero', referring to the count of non-zero elements in the coefficient matrix of the MIP problems.
>
>
> **Reference**
>
> [1] Bowly, S., Smith-Miles, K., Baatar, D., & Mittelmann, H. (2020). Generation techniques for linear programming instances with controllable properties. _Mathematical Programming Computation_, _12_(3), 389-415.
>
> [2] https://miplib.zib.de/tag_benchmark.html

---

### Official Review · Reviewer_Y5jX · 2023-11-07

**Soundness:** 3 good
**Presentation:** 2 fair
**Contribution:** 2 fair
**Rating:** 3
**Confidence:** 2

**Summary:**

The paper provides an MIP instance generation method. Bipartite graph embedding is used with VGAE and EM algorithms for instance classification. Node splitting and merging are used to generate isomorphic MIP problems, which is also applicable to generate scalable MIP instances. Experiments are conducted to assess the graph modularity and optimality gap, indicating the preservation of MIP properties by the generation method.

**Strengths:**

It is an interesting research to generate similar instances given limited instances. The augmented instances can be used to assist in the machine learning framework which needs considerable data for training. The proposed techniques borrow the current bipartite graph neural network, GVAE and self-supervised learning, which are sound for learning representations of instances.

**Weaknesses:**

The description is not very good in this paper with some parts not explained well. The definition of graph modularity is not given and commonly used metrics for fidelity are not introduced. The infeasibility of instances are not well addressed. It is not clear what ML-based framework is used in experiments to evaluate the  MIPGen and how it works. The details are important to get insights on the effect of the generated data in improving the ML models.

Additionally, the experiments are not enough to verify the significance of MIPGen. First, only graph modularity may not be sufficient to gauge the graph fidelity and more metrics are necessary to ensure a comprehensive similarity comparison. Second, the whole MIPGen are complex with many heavy modules. The simple instance generation ought to be compared to underscore the effect of MIPGen. For example, simply fitting the degrees and parameters in instances into a distribution that may be already enough to generate new instances.

The key techniques in this paper are not novel. The bipartite graph neural network still follows an early work, which can not well differentiate the integer and continuous variables. Moreover, there is no MIP but only IP instances in MIS, CA, MVC. GVAE and EM with GMM are also not new. It would be more convincing if authors show the intermediate outputs in the MIPGen like the clustered instances by EM and latent representations learned by GVAE, to showcase the effects of the chosen techniques.

**Questions:**

1. Given the excerpt "As a lossless representation for MIP instances (Gasse et al., 2019), Bipartite Graph Representation (BGR) seamlessly transforms MIP instances into bipartite graphs...", can authors explain why the bipartite graph embedding in (Gasse et al., 2019) is lossless? Is there any proof for that?
2. During node merging, the authors claim "the feature set $A_i$ or $B_j$ of the newly merged node is randomly selected from one of the two nodes being merged." Can random selected features cause infeasible instances? I didn't see any description about how the infeasibility is avoided.
3. In section 4.1.3, only 20 instances can increase the performance a lot. Does it mean the original training dataset is small? If so, how to train the heavy modules in MIPGen like GVAE, BGR? On the contrary, if the training instances are sufficient, is MIPGen still useful?

---

> ### Author Response · Authors · 2023-11-23
> **Reply to reviewer Y5jX (1/2)**
>
> Thank you for your constructive comments and suggestions, which have been exceedingly helpful in improving our paper. We have carefully incorporated them into the revised manuscript. Below, we restate your comments followed by our detailed responses:
>
> >1. The definition of graph modularity is not given and commonly used metrics for fidelity are not introduced.
> >2. Additionally, the experiments are not enough to verify the significance of MIPGen. First, only graph modularity may not be sufficient to gauge the graph fidelity and more metrics are necessary to ensure a comprehensive similarity comparison.
>
> We appreciate your insightful suggestion! We have now included additional experiments focusing on problem similarity. Alongside the original structural evaluation metrics, we have introduced new metrics such as coefficient density, node degree distribution, and the distribution of LHS/RHS. These metrics further assess the similarity between newly generated and original problems. For detailed information, please refer to section 4.1.2 of the updated paper.
>
> >3. The infeasibility of instances are not well addressed.
>
> Our paper's primary innovation lies in expanding the scale of generated problems. To achieve this objective, we implemented a split-merge strategy. While this approach, involving random splitting and merging, may lead to the creation of infeasible instances, it's noteworthy that all instances generated for this study were feasible. We recognize the potential for generating infeasible instances and intend to incorporate MILP duality in our future research to ensure the feasibility of generated problems. Additionally, it's important to note that infeasible problems can also be valuable. For instance, the MIPLIB benchmark includes several infeasible problems which require solvers to spend a considerable amount of time to determine their infeasibility. Your insightful advice is greatly appreciated.
>
> >4. Given the excerpt "As a lossless representation for MIP instances (Gasse et al., 2019), Bipartite Graph Representation (BGR) seamlessly transforms MIP instances into bipartite graphs...", can authors explain why the bipartite graph embedding in (Gasse et al., 2019) is lossless? Is there any proof for that?
>
> The term "lossless" in this context means that the bipartite graph representation retains all information contained in a MIP problem. This representation is lossless because the nodes on each side of the bipartite graph encapsulate all information of the variables and constraints of the MIP problem, while the edges represent the coefficient matrix.
>
> >5. It is not clear what ML-based framework is used in experiments to evaluate the MIPGen and how it works.
>
> As mentioned in section 4.1.3 of the paper, we utilized the framework described in [1] for our data augmentation experiments related to ML-based solutions.
>
> >6. The simple instance generation ought to be compared to underscore the effect of MIPGen. For example, simply fitting the degrees and parameters in instances into a distribution that may be already enough to generate new instances.
>
> Thank you for your recommendation. We have included a baseline comparison with the method described in [3]. Our approach has been contrasted with this baseline in terms of problem-solving difficulty, problem similarity, and data augmentation experiments. For more details, please refer to section 4 of the updated paper.

---

> > ### Author Response · Authors · 2023-11-23
> > **Reply to reviewer Y5jX (2/2)**
> >
> > >7. The key techniques in this paper are not novel. The bipartite graph neural network still follows an early work, which can not well differentiate the integer and continuous variables.
> >
> > Building upon the work in [2], we have introduced a novel split-merge framework for generating MIP problems, complemented by data augmentation experiments to validate our method's effectiveness. Additionally, MIPGen can integrate tree-like structures to generate larger-scale MIP problems, enabling ML-based frameworks trained on smaller problems to solve larger-scale instances of the same type. We aim to explore more complex generation strategies based on the split-merge framework in our future work.
> >
> > >8. There is no MIP but only IP instances in MIS, CA, MVC.
> >
> > IP problems are a specific subset of MIP problems. Our paper focuses on generating experiments with IP problems, but our methodology is also applicable to generating MIP problems.
> >
> > >9. It would be more convincing if authors show the intermediate outputs in the MIPGen like the clustered instances by EM and latent representations learned by GVAE, to showcase the effects of the chosen techniques.
> >
> > In the revised version of our paper, specifically in section xx, figure x, we have included diagrams illustrating the intermediate clustering results.
> >
> > >10. In section 4.1.3, only 20 instances can increase the performance a lot. Does it mean the original training dataset is small? If so, how to train the heavy modules in MIPGen like GVAE, BGR? On the contrary, if the training instances are sufficient, is MIPGen still useful?
> >
> > The motivation behind the data augmentation experiments in MIPGen is to address the issue of insufficient training data in ML-based frameworks. In these experiments, we focused on enhancing datasets with a limited number of problem instances, hence the original training dataset is indeed small. The relationship between the ML-based framework and the amount of training data is pivotal; even with a single MIP problem instance, we were able to achieve significant improvements (5-10% increase) after data augmentation. The current quantity of training datasets suffices for effective training outcomes. Nevertheless, even with ample training data, generating larger, varied datasets is crucial to enhance the generalizability of ML-based frameworks to solve MIP problems of different scales and complexities.
> >
> >
> > **Reference**
> >
> > [1] Ye, H., Xu, H., Wang, H., Wang, C., & Jiang, Y. (2023). GNN&GBDT-Guided Fast Optimizing Framework for Large-scale Integer Programming. ICML 2023
> >
> > [2] You, J., Wu, H., Barrett, C., Ramanujan, R., & Leskovec, J. (2019). G2SAT: Learning to generate SAT formulas. _Advances in neural information processing systems_, _32_.
> >
> > [3] Bowly, S., Smith-Miles, K., Baatar, D., & Mittelmann, H. (2020). Generation techniques for linear programming instances with controllable properties. _Mathematical Programming Computation_, _12_(3), 389-415.

---

### Official Review · Reviewer_si4d · 2023-11-11

**Soundness:** 2 fair
**Presentation:** 2 fair
**Contribution:** 2 fair
**Rating:** 3
**Confidence:** 4

**Summary:**

This paper proposes a MIP instance generation method. It first clusters the MIPs, and applies the node splitting-and-merging process. It also proposes to concatenate tree structures to get larger problems. Experiments show that the generated MIPs are similar to the original ones and can enhance the solution effect of the ML-based framework.

**Strengths:**

1. This paper successfully applies the split-and-merge process, which was originally for generating SATs, for generating MIPs.
1. It can control the size of the new MIP instances.

**Weaknesses:**

1. The method is an application of existing SAT generation method into MIPs. The novelty design for MIPs is not sufficient.
1. In the split-and-merge process, the constraints and varaiables features are not newly generated. So it is more appropiate for generating SATs instead of MIPs.
1. The considered graph statistics are only for structures, but not for coefficients. So the authors should consider new metrics for MIPs instead of SATs.
1. The authors may want to add some baselines to demonstrate the effectiveness.
1. A recent work [1], which appears in NeurIPS2023, also uses deep learning to generate MIPs. So this paper is not the first one for this topic.


[1] [https://arxiv.org/pdf/2310.02807.pdf](https://arxiv.org/pdf/2310.02807.pdf)

**Questions:**

1. How to generate new node features and edge coefficients?
1. Among the considered three datasets, MIS has all coefficients as 1, so it can be learned by the split-and-merge method like SAT problems. Do the other datasets have this characteristic?
1. How long is the fixed wall time in Table 1? What do not report the solving time of the instances?
1. What is the meaning of the reconstruction percentage?

---

> ### Author Response · Authors · 2023-11-23
> **Reply to reviewer si4d (1/2)**
>
> Thank you for your constructive comments and suggestions, and they are exceedingly helpful for us to improve our paper. We have carefully incorporated them in the revised paper. In the following, your comments are first stated and then followed by our point-by-point responses.
>
> >1. The method is an application of existing SAT generation method into MIPs. The novelty design for MIPs is not sufficient.
>
> Building upon the work in [2], we introduced the split-merge framework to the generation of MIP problems, complemented by data augmentation experiments to validate the efficacy of our method. Furthermore, MIPGen can amalgamate tree-like structures to generate larger-scale MIP problems. This enables the training of ML-based frameworks on smaller MIP problems to solve larger-scale problems of the same type. We plan to explore more complex generation strategies based on the split-merge framework in future work.
>
> >2. In the split-and-merge process, the constraints and variables features are not newly generated. So it is more appropriate for generating SATs instead of MIPs.
> >3. How to generate new node features and edge coefficients?
>
> While our current method does not generate new features, it effectively rearranges the bipartite graph structure and splices tree-like structures. This approach perturbs the coefficient matrix and RHS of MIP problems while maintaining characteristics similar to the original problem, thus generating MIP problems akin to those being learned. We will consider the generation of new features in our subsequent work.
>
> >4. The considered graph statistics are only for structures, but not for coefficients. So the authors should consider new metrics for MIPs instead of SATs.
>
> We appreciate your suggestion! We have added experiments on problem similarity. In addition to the original structural evaluation metrics, we have now employed additional metrics, including coefficient density, node degree distribution, distribution of LHS/RHS to assess the similarity between the newly generated and the original problems. We refer to section 4.1.2 of updated paper for detailed information.
>
> >5. The authors may want to add some baselines to demonstrate the effectiveness.
>
> Thank you for your recommendation. We have included the baseline from [3] for comparison. Our method has been compared with this baseline in terms of problem-solving difficulty, problem similarity, and data augmentation experiments. Please refer to section 4 of the updated paper for more details.
>
> >6. A recent work [1], which appears in NeurIPS 2023, also uses deep learning to generate MIPs. So this paper is not the first one for this topic.
>
> Acknowledging the work in [1], MIPGen is indeed not the first to use deep learning for generating MIP problems. However, it's important to note that this paper was released on ArXiv on 2023.10.4, which is after the ICLR submission deadline (2023.9.28). We have adjusted our statements accordingly in the revised paper.
>
> >7. Among the considered three datasets, MIS has all coefficients as 1, so it can be learned by the split-and-merge method like SAT problems. Do the other datasets have this characteristic?
>
> The datasets we used, namely MIS, CA, and MVC, all have edge weights set to 1. Nevertheless, MIPGen is capable of handling problems with varying edge weights, as the edge weights in the bipartite graph are assigned corresponding coefficients. Additionally, the GNN model used by MIPGen is edge feature-aware.

---

> > ### Author Response · Authors · 2023-11-23
> > **Reply to reviewer si4d (2/2)**
> >
> > >8. How long is the fixed wall time in Table 1? What do not report the solving time of the instances?
> >
> > Thank you for pointing this out. In Table 1, the fixed wall time is set at 1200s, which we have now included in our revised paper. The reason for not providing complete solving times in our experimental data is due to the selection of benchmark problems like the MIS problem, which are NP-hard and cannot be optimally solved in a short time. In our trials using SCIP to solve MIS problems from our dataset (10,000 variables, 30,000 constraints), we did not obtain an optimal solution even after 24 hours. Therefore, we chose a relatively reasonable wall time of 1200s to compare the optimal gap and demonstrate the problem-solving difficulty.
> >
> > >9. What is the meaning of the reconstruction percentage?
> >
> > Let $n$ be the number of nodes in the bipartite graph, and $m$ the number of edges. If MIPGen performs $x$ node splitting and merging operations to generate a new problem, we define the reconstruction percentage as $\frac{x}{m-n}$. We have added a detailed explanation of this in our revised paper.
> >
> >
> > **Reference**
> >
> > [1] Geng, Z., Li, X., Wang, J., Li, X., Zhang, Y., & Wu, F. (2023). A deep instance generative framework for milp solvers under limited data availability. _arXiv preprint arXiv:2310.02807_.
> >
> > [2] You, J., Wu, H., Barrett, C., Ramanujan, R., & Leskovec, J. (2019). G2SAT: Learning to generate SAT formulas. _Advances in neural information processing systems_, _32_.
> >
> > [3] Bowly, S., Smith-Miles, K., Baatar, D., & Mittelmann, H. (2020). Generation techniques for linear programming instances with controllable properties. _Mathematical Programming Computation_, _12_(3), 389-415.

---

### Meta-Review · Area_Chair_t5tz · 2023-12-11

**Metareview:**

This paper introduces a learning-based approach for generating MIP instances, which can be beneficial for training machine learning-based MIP solving policies. The approach uses a bipartite graph representation to learn to cluster the training instances. It then applies a node splitting-and-merging process to the clusters. The approach also proposes to concatenate tree structures to create larger problems. Experiments show that the generated instances are similar to the training instances.

Strengths: First, it is indeed an important problem to generate sufficient training data for training practical deep learning based MIP solving techniques. This paper successfully applies the split-and-merge process, which was originally designed for generating SATs, for generating MIPs. It can control the size of the new MIP instances.

Weaknesses: Reviewer X6GL brought up valid points regarding weaknesses #2 and #3. In their response to weakness #2, the authors stated that “3 include statistical indicators of MIP problems, such as the percentage of binary (integer or continuous) variables, a vector of coefficients of the objective function, and a vector representing the number of constraints for each variable ii. However, this method lacks a representation of the key structural aspect of the MIP problem's coefficient matrix. We believe that basic statistical indicators are insufficient to distinguish complex MIP problems encountered in the real world.” While the authors believe that basic statistical indicators are insufficient to distinguish complex MIP problems encountered in the real world, it would be beneficial to compare their method to such indicators as a simple baseline, as suggested by the reviewer. Additionally, weakness #3 states that “The experiments are small-scale. For example, in Table 2, only three instances are considered, each is augmented to 20 instances.” This paper would benefit from having larger-scale experiments.

Weaknesses #2 and #3 brought up by reviewer si4d are also valid. The authors’ response acknowledged that these are indeed weaknesses of the paper.

**Justification For Why Not Higher Score:**

See the weaknesses.

**Justification For Why Not Lower Score:**

N/A

---

### Decision · Program_Chairs · 2024-01-16

Reject